# Differential Gene Regulatory Network Analysis between Azacitidine-Sensitive and -Resistant Cell Lines

**DOI:** 10.3390/ijms25063302

**Published:** 2024-03-14

**Authors:** Heewon Park, Satoru Miyano

**Affiliations:** 1School of Mathematics, Statistics and Data Science, Sungshin Women’s University, Seoul 02844, Republic of Korea; 2M&D Data Science Center, Tokyo Medical and Dental University, 1-5-45 Yushima, Bunkyo-ku, Tokyo 113-8510, Japan; miyano@hgc.jp; 3Human Genome Center, Institute of Medical Science, University of Tokyo, 4-6-1 Shirokane-dai, Minato-ku, Tokyo 108-8639, Japan

**Keywords:** molecular interplay, acute myeloid leukemia, anti-cancer drug resistance, metallothionein gene family

## Abstract

Azacitidine, a DNA methylation inhibitor, is employed for the treatment of acute myeloid leukemia (AML). However, drug resistance remains a major challenge for effective azacitidine chemotherapy, though several studies have attempted to uncover the mechanisms of azacitidine resistance. With the aim to identify the mechanisms underlying acquired azacitidine resistance in cancer cell lines, we developed a computational strategy that can identify differentially regulated gene networks between drug-sensitive and -resistant cell lines by extending the existing method, differentially coexpressed gene sets (DiffCoEx). The technique specifically focuses on cell line-specific gene network analysis. We applied our method to gene networks specific to azacitidine sensitivity and identified differentially regulated gene networks between azacitidine-sensitive and -resistant cell lines. The molecular interplay between the metallothionein gene family, C19orf33, ELF3, GRB7, IL18, NRN1, and RBM47 were identified as differentially regulated gene network in drug resistant cell lines. The biological mechanisms associated with azacitidine and AML for the markers in the identified networks were verified through the literature. Our results suggest that controlling the identified genes (e.g., the metallothionein gene family) and “cellular response”-related pathways (“cellular response to zinc ion”, “cellular response to copper ion”, and “cellular response to cadmium ion”, where the enriched functional-related genes are MT2A, MT1F, MT1G, and MT1E) may provide crucial clues to address azacitidine resistance in patients with AML. We expect that our strategy will be a useful tool to uncover patient-specific molecular interplay that provides crucial clues for precision medicine in not only gastric cancer but also complex diseases.

## 1. Introduction

Acute myeloid leukemia (AML), a type of blood cancer, affects the bone marrow and progresses rapidly if untreated. During the last two decades, hypomethylating agents including azacitidine have been the mainstay of treatment for AML [1]. In particular, azacitidine is the recommended front-line treatment for older patients with AML [2]. However, azacitidine chemotherapy is not always effective due to the drug resistance, which threatens patient survival. Numerous studies have attempted to uncover the mechanisms of azacitidine resistance in AML [3,4,5,6]. Sripayap et al. [3] generated two azacitidine-resistant cell lines and uncovered genetic disparities between the resistant cell lines. They revealed that the resistance arises from negating azacitidine-mediated activation of apoptosis signaling and reestablishing G2/M checkpoint. Minařík et al. [6] developed a model of 5-azacytidine resistance from myelodysplastic syndromes (MDS) /AML cell lines and observed the deregulation of several cancer-related pathways including the phosphatidylinosito-3 kinase signaling when investigating mRNA expression and DNA variants of the azacytidine resistant phenotype.

Although many previous studies were conducted to uncover the azacitidine resistance mechanism based on differential gene expression analysis [6,7], the complex mechanisms of drug resistance cannot be understood by perturbation of a single gene, because the drug resistance is caused by abnormalities in complex molecular networks, which should be considered. The mechanisms underlying acquired drug resistance were poorly understood, making drug resistance a major problem.

Recently, gene regulatory network analysis has become one of the most powerful strategies to uncover the biological mechanisms of diseases and has been widely used in studies on the identification of complex disease systems and drug-response genes, etc. [8,9,10,11]. In this study, we aimed to uncover the mechanisms of acquired azacitidine resistance based on gene regulatory network analysis, especially in a cell line-specific gene network. In general, the gene regulatory network consists of more than 10,000 genes and has a considerably complex structure with a huge number of edges. Thus, interpretation of the network and marker identification based on a complex gene regulatory network remains a challenge. The existing studies have focused on the well-known markers and then uncovered the molecular interplay surrounding the markers, because the comprehensive analysis of the huge and complex gene regulatory networks was impossible. We consider differentially regulated gene network identification to be one of the approaches for interpreting a complex gene network. Recently, various computational methods for differential gene network analysis have been developed for a range of research purposes. Grimes et al. [12] developed a method for differential network analysis that measures dissimilarity between networks based on the differential connectivity of a set. Tu et al. [13] also developed a method to measure dissimilarity between gene networks that simultaneously considers the changes in gene interactions and changes in expression levels. Xie et al. [14] proposed a differential network flow method that measures the distribution differences in network flow for each gene in two networks. A computational strategy to detect a statistically significant network was developed, where the generalized Hamming distance was used to evaluate the topological difference between networks [15]. Park et al. [16] developed a statistical method for differentially regulated gene network identification based on comprehensive information of the gene network structure and extended the method to cell line characteristic-specific gene network analysis.

To reveal the crucial markers and their molecular interplay that characterize azacitidine resistance cell lines, we extend DiffCoEx for cell line characteristic-specific gene network analysis and developed a computational strategy for differential gene network identification based on the topological overlap. The developed method measures the dissimilarity network based on a statistic computed with an edge size and node similarity (neighbors of genes) based on the topological overlap, while the CIdrgn [16] is constructed using the sum of several statistics; thus, normalization should be performed, because the statistics have different scales. We applied our method to the publicly accessible DepMap database (https://depmap.org/portal/ (accessed on 4 August 2022)) and estimated the azacitidine sensitivity-specific gene networks that vary according to the drug sensitivity values of cell lines. We then identified the differentially regulated gene networks between drug-sensitive and -resistant cell lines. The gene network comprising RBM47, ELF3, GRB7, and the strong suppression of NRN1 by C19orf33 were identified as azacitidine resistance-specific molecular interactions. Among the identified networks, the interplay between the metallothionein gene family was observed in both drug-sensitive and -resistant cell lines, while the network showed different edge strength between drug-sensitive and -resistant cell lines. The pathways related to “cellular responses” (i.e., “cellular response to metal ion”, “cellular response to zinc ion, “ cellular response to copper ion”, and “cellular response to cadmium ion ”, where the enriched functional-related genes are MT2A, MT1F, MT1G, and MT1E) were identified as enriched pathways related to the differentially regulated gene network by using GO term pathway analysis. We validated the identified markers through a literature survey and found that more than half of the identified genes were considered as markers of azacitidine and AML.

The conclusions of our analysis are that *1. the gene network between RBM47, ELF3, and GRB7 is an azacitidine resistance-specific characteristic; 2. the molecular interplay of the metallothionein gene family is a crucial marker for uncovering the mechanism of azacitidine sensitivity in cancer cell lines.* Our results suggest that suppression and/or activation of the identified genes and “cellular response”-related pathways may be a key to uncovering the mechanisms of acquired azacitidine resistance in patients with AML.

## 2. Results

### Differentially Regulated Gene Network Identification between
Azacitidine-Resistant and -Sensitive Cell Lines

We applied the developed method, which will be introduced in Section 4, for differentially regulated gene network identification on the DepMap database (https://depmap.org/portal/ (accessed on 04 Aug 2022)). The expression dataset (i.e., CCLE_expression.csv) of mRNA expression levels comprises 19,221 genes for 1406 cell lines. The drug sensitivity (primary replicate collapsed log fold change) of azacitidine in 578 cell lines was used for the estimation of azacitidine sensitivity-specific gene networks. We focused on the 1000 genes with the highest variance in expression levels and 549 cell lines with non-missing values in both expression level and drug sensitivity datasets. For the 549 cell lines corresponding to the azacitidine sensitivity values $m_{\alpha },\alpha =1,...,549$, we applied NetworkProfiler, a machine learning strategy for cell line-specific gene network analysis (see Section 4) and estimated that 549 gene regulatory networks varied depending on the azacitidine sensitivity values of cell lines.

We defined 100 azacitidine-sensitive and -resistant cell lines corresponding to the 100 largest and smallest values of the drug sensitivity, respectively. We focus on azacitidine resistance-specific gene networks and extracted edges corresponding to the top 1% largest absolute edge size from the estimated 100 drug-resistant cell line-specific gene networks.

The edges comprised 32 subnetworks. For the 32 subnetworks, we applied our method to reveal differentially regulated gene networks between azacitidine-sensitive and -resistant cell lines, where three subnetworks were identified.

Figure 1 shows the identified gene networks, where the top and bottom show their molecular interplay in drug resistance and sensitive cell lines, respectively.

As shown in Figure 1, azacitidine-sensitive and -resistant cell lines showed distinguishing gene regulatory networks, i.e., drug-resistant cell lines have relatively dense gene networks compared with drug-sensitive cell lines. The gene network of the metallothionein gene family (i.e., MT1E, MT1F, MT1G, and MT2A) was observed in both drug-sensitive and -resistant cell lines, whereas the strength of the interplay becomes weaker as the sensitivity of the cell line increases. RBM47 strongly regulates ELF3 in azacitidine-resistant cell lines, whereas the subnetwork of RBM47, ELF3, and GRB7 disappeared in drug-sensitive cell lines. Furthermore, the suppression of NRN1 by C19orf33 also disappeared in the sensitive cell lines. The genes (i.e., metallothionein gene family, C19orf33, ELF3, GRB7, IL18, NRN1, and RBM47) in the identified differentially regulated gene networks can be considered as key markers to understand the mechanisms involving drug resistance of azacitidine and AML. Table 1 shows the identified markers and previous studies on their mechanisms related to azacitidine and AML.

As shown in Table 1, more than half of the identified genes were studied as markers of not only azacitidine but also AML.


**ELF3**
Li et al. [18] identified ELF3 as one of the 5-azacytidine immune genes and ELF3 was classified as the gene set category of “Inflammation”. It was suggested that ELF3-AS1 could be a prognostic factor and influences the prognosis of AML by Guo et al [17].
**IL18**
Saadi et al. [19] evaluated the expression levels of IL18 in AML patients according to their response to treatment and showed that expression levels of IL18 were increased in AML patients who did not respond to therapy compared to those patients who respond to therapy. Furthermore, it was observed that the expression levels of IL18 were significantly increased in high-risk groups of AML patients [19]. A correlation was observed between the levels of IL18 and the prognosis of AML, i.e., higher levels of IL18 were correlated with worse prognosis of AML [20]. The expression profiles of IL-6 and IL18 were considered as prognostic markers for AML [19]. Furthermore, the higher expression levels of IL18 and its receptor induced drug resistance in AML [19]. From the results, Saadi et al. [19] suggested that IL18 is an important prognostic marker in AML and control of the expression and regulation of IL18 may play key roles in the management of AML. Chen et al. [21] showed that overexpression of IL18 might reflect the convergence of several important unfavorable prognostic factors in AML. Song et al. [22] also suggested that high circulating levels of IL18 are a potential predictor for a decreased risk of AML. It was also demonstrated that the variant GT genotype of IL18 rs1946518 led to poorer survival rates in AML [23]. 
**Metallothionein (MT) gene family**
Exposure of cells in culture to 5-azacytidine stimulates the expression of the metallothionein gene [26]. Stallings et al. [28] showed that 5-azacytidine-induced conversion to cadmium resistance is correlated with early S-phase replication of inactive metallothionein genes in synchronized cadmium-sensitive cells. Increased MT-I expression is a poor prognostic marker for AML [24]. MT was identified as a drug-resistance-related protein and was expressed in leukemic cells in more than half of cases of newly diagnosed AML [24]. Patricia et al. [25] showed that MT1 is critical for the growth and survival of DNMT3A;NPM1-mutant AML cells and nominated MT1 as a key marker for the treatment and prevention of DNMT3A;NPM1-mutant AML.
**RBM47**
5-azacytidine-resistant metabolic adaptable cells have several other alterations in RBM47 expression [31]. RBM47 was identified as an important AML-specific RBP gene, and Saha et al. [30] revealed that RBM47 is a potential candidate for therapeutic intervention toward effective eradication of leukemic stem cells in AML.

It can be seen through the literature survey that our analysis provided biologically reliable results for identifying crucial markers and for understanding the molecular interplay that has the potential to improve the therapeutic efficiency of azacitidine in patients with AML. Although some genes were not yet identified as a marker, the genes and their interplay can be considered as novel markers to understand the mechanism of azacitidine resistance in cancer cell lines.

To identify the biological pathways involved in the differentially regulated gene networks in azacytidine-resistant cell lines, we performed a Gene Ontology (GO) term pathway analysis of the genes in the identified network. Figure 2 shows the significantly enriched pathways corresponding to the *p*.value (i.e., −log(*p*.value)).

As shown in Figure 2, “GO:0071248: cellular response to metal ion (definition: *Any process that results in a change in state or activity of a cell (in terms of movement, secretion, enzyme production, gene expression, etc.) as a result of a metal ion stimulus.*)” was the most enriched pathway. Furthermore, “cellular response”-related pathways (“GO:0071294: cellular response to zinc ion (definition: *Any process that results in a change in state or activity of a cell (in terms of movement, secretion, enzyme production, gene expression, etc.) as a result of a zinc ion stimulus*)”, “GO:0071280: cellular response to copper ion (definition: *Any process that results in a change in state or activity of a cell (in terms of movement, secretion, enzyme production, gene expression, etc.) as a result of a copper ion stimulus*)”, “GO:0071276: cellular response to cadmium ion (definition: *Any process that results in a change in state or activity of a cell (in terms of movement, secretion, enzyme production, gene expression, etc.) as a result of a cadmium (Cd) ion stimulus*)”, where the enriched functional-related genes are MT2A, MT1F, MT1G, and MT1E), were identified as crucial biological pathways for azacitidine resistance-specific molecular interplay. The cellular response is a reaction of a cell to extracellular signals. “GO:0071276: cellular response to cadmium ion” was identified as enriched pathway for genes that show different expression levels between AML subgroups and normal bone marrow (BM) [32]. The mechanisms of cellular responses involving resistance to chemotherapy in cancer have been demonstrated in several studies [33,34,35]. Cheng et al. [33] suggested that drug efficacy can be enhanced by Asplatin through altering the cellular response.

We suggest though our results and literature survey that the metallothionein gene family and the enriched pathways associated with them (i.e., “cellular response”-related pathways) may play crucial roles in acquisition of azacitidine resistance of cancer cells. Suppression and/or activation of the genes in the identified network and their interplay may provide crucial insights to understand and address azacitidine resistance of cancer cell lines.

## 3. Discussion

We aimed to uncover the mechanisms of acquired azacitidine resistance in cancer cell lines based on gene network analysis. In particular, we focused on cell line-specific gene regulatory networks that vary depending on the characteristics of cell lines.

To uncover the azacitidine resistance mechanism, we developed a computational method for differentially regulated gene network identification by development of a computational strategy that can identify differentially regulated gene networks between drug-sensitive and -resistant cell lines by extending the existing method, i.e., DiffCoEx.

We first estimated azacitidine sensitivity-specific gene regulatory networks by using the NetworkProfiler, which is the methodology for cell line-specific gene network estimation. We then applied the developed method to the estimated 549 gene networks’ corresponding 549 azacitidine sensitivity values and revealed azacitidine resistance-specific molecular interplay. As a result, three differentially regulated gene networks were extracted, where the metallothionein gene family, C19orf33, ELF3, GRB7, IL18, NRN1, and RBM47 were identified as members of the identified networks. The subnetwork comprising RBM47, ELF3, and GRB7 was found to be an azacitidine resistance-specific gene regulatory network. The strong suppression of NRN1 by C19orf33 was also considered a drug resistance-specific molecular interplay. Among the identified three subnetworks, the interplay between the metallothionein gene family (i.e., MT1E, MT1F, MT1G, and MT2A) was observed in drug-sensitive and -resistant cell lines, and their interplay becomes weaker with an increase in the drug sensitivity of a given cell line. The “cellular response”-related pathways were identified as enriched pathways of the genes in the differentially regulated gene network, where the enriched function-related genes were in the metallothionein gene family.

Our findings suggest that suppressing and/or activating the identified genes and “cellular response”-related pathways may provide crucial information to help understand the mechanisms of acquired azacitidine resistance in cancer cell lines.

Although our strategy and results were validated by literature, the statistical accuracy for the differential gene network analysis is also considered to validate the proposed method. Further work remains to be performed on the evaluation of our method through comparison with other methods.

## 4. Method for Differential Gene Network Analysis

In this section, we introduce the computational method for azacitidine sensitivity-specific gene regulatory network estimation and differentially regulated gene network identification.

Let $X={\left(x_{1},...,x_{n}\right)}^{T}\in R^{n\times p}$, where $x_{i}={\left(x_{i1},...,x_{ip}\right)}^{T}$ is the expression levels of *p* regulator genes and $y_{l}\in R^{n}$ is the expression levels of the $l^{th}$ target gene. We suppose that the *p* regulator genes control the target gene transcription for *n* cell lines. We consider the following linear regression model to represent the gene regulatory network,
(1)$$\begin{array}{c} y_{il}=\beta_{l}^{T}x_{i}+\epsilon_{il},\;i=1,...,n,\;l=1,...,q, \end{array}$$
where $\beta_{l}={\left(\beta_{l1},...,\beta_{lp}\right)}^{T}$ is the regression coefficient vector that indicates the effect of *p* regulator genes on the $l^{th}$ target gene, and $\epsilon_{il}$ is a random error vector for the model of the $l^{th}$ target gene. The regulatory strength of the *p* regulatory genes on the $l^{th}$ target gene is described by the estimated regression coefficient $\beta_{l}$. To estimate the regression coefficient $\beta_{l}$, various statistical methods have been developed and used in various fields of research. We consider the $L_{1}$-type regularization methods (e.g., lasso [36], elastic net [37]) that perform gene selection and edge strength estimation simultaneously. In particular, the elastic net has been often used to gene network analysis, because it enables us to select more than *n* (e.g., number of cell lines) genes.

Although the regression model in (Equation 1) has been frequently used to represent gene regulatory networks and is easy to understand, the model cannot describe cell line-specific molecular interactions because the regulatory effect of gene *p*, i.e., $\beta_{l}$, is given for all *n* cell lines, i.e., cell line-specific $\beta_{l}$ cannot be described by the model. Figure 3 shows the correlation between two genes (i.e., UPF1 and DPM1) in azacitidine-sensitive, moderately azacitidine-sensitive, and azacitidine-resistant cell lines. As shown in Figure 3, DUSP23 and AKR1C3 show a positive correlation in azacitidine-resistant cell lines (bottom right of Figure 3), whereas the positive interplay disappeared in drug-sensitive cell lines (top right of Figure 3). In other words, the interplay between DUSP23 and AKR1C3 shows distinct patterns in drug-sensitive, moderately drug-sensitive, and drug-resistant cell lines. Furthermore, the scatter plot for all cell lines (top left of Figure 3) cannot accurately describe the pattern of their correlation, which vary depending on the drug sensitivity. These findings indicate that gene regulatory networks should be constructed after considering cell line characteristics (i.e., azacitidine sensitivity of cell lines).

### 4.1. Azacitidine Sensitivity-Specific Gene Regulatory Network Estimation

To estimate azacitidine sensitivity-specific gene regulatory networks which vary according to the drug sensitivity of cell lines, we consider the following varying coefficient model [38]
(2)$$\begin{array}{c} y_{il}=\beta_{l}^{T}\left(m_{\alpha }\right)x_{i}+\varepsilon_{il}, \end{array}$$
where $\beta_{l}\left(m_{\alpha }\right)={\left(\beta_{l1}\left(m_{\alpha }\right),...,\beta_{lp}\left(m_{\alpha }\right)\right)}^{T}$ is the varying coefficient vector that describes the effects of *p* regulatory genes on the $l^{th}$ target gene in the $\alpha^{th}$ cell line having the azacitidine sensitivity value $m_{\alpha }$. The varying coefficient model allows us to describe the azacitidine sensitivity value $m_{\alpha }$-specific gene regulatory network, that is, we can estimate *n* gene regulatory networks for *n* cell lines corresponding to *n* azacitidine sensitivity values ($m_{\alpha }$, α=1,...,n). Thus, we can reveal the molecular interplay changes according to the sensitivity to azacitidine and identify key markers and pathways involved in the mechanisms of acquired azacitidine resistance.

The varying coefficient $\beta_{l}\left(m_{\alpha }\right)$ was estimated by the following kernel-based $L_{1}$ type regularization, called a NetworkProfiler [39],
(3)$$\begin{array}{c} {\hat{\beta }}_{l}\left(m_{\alpha }\right)=\underset{\beta_{l}\left(m_{\alpha }\right)}{\arg\;\min}\left\{\frac{1}{2}\sum_{i=1}^{n}{\left\{y_{il}-\beta_{l}^{T}\left(m_{\alpha }\right)x_{i}\right\}}^{2}K\left(m_{i}-\;m_{\alpha }|b_{l}\right)+P\left\{\beta_{l}\left(m_{\alpha }\right)\right\}\right\}, \end{array}$$
where $P\{\beta_{l}\left(m_{\alpha }\right)\}$ is the recursive elastic net penalty with regularization parameters $\lambda_{l\alpha }>0$ and $0<\delta_{l\alpha }<1$,
(4)$$\begin{array}{c} P\left\{\beta_{l}\left(m_{\alpha }\right)\right\}=\lambda_{l\alpha }\sum_{j=1}^{p}[\frac{1}{2}\left(1-\delta_{l\alpha }\right)\beta_{lj\alpha }^{2}+\delta_{l\alpha }w_{lj\alpha }|\beta_{lj\alpha }\left|\right], \end{array}$$
and
(5)$$\begin{array}{c} K\left(m_{i}-m_{\alpha }|b_{l}\right)=\exp\left\{\frac{-{\left(m_{i}-m_{\alpha }\right)}^{2}}{b_{l}}\right\}, \end{array}$$
is the Gaussian kernel function, where $b_{l}$ is a bandwidth to control width of kernel function. In the NetworkProfiler, the Gaussian kernel function plays a key role to group cell lines according to the characteristics of cell lines (i.e., azacitidine sensitivity $m_{i}$). When we estimate the azacitidine sensitivity value $m_{\alpha }$-specific gene network, we measure similarity between drug sensitivity values of cell lines, i.e., $m_{\alpha }$ and $m_{i}$ for i=1,...,n, based on ${\left(m_{i}-m_{\alpha }\right)}^{2}$, then determine weights of each cell lines that control the influence of cell lines in estimating the azacitidine sensitivity value $m_{\alpha }$-specific gene network. In other words, the Gaussian kernel function $K(m_{i}-m_{\alpha }|b_{l})$ is used as weight to control the influence of cell lines and enables us to estimate $m_{\alpha }$-specific gene networks based only on cell lines with a similar characteristic $m_{i}$ to that of the target cell lines, i.e., $m_{\alpha }$. This implies that the NetworkProfiler can identify the specific molecular interactions for a cancer related status of cell lines, i.e., $m_{\alpha }$.

The NetworkProfiler is implemented by the open-source software SiGN-L1 of the Super Computer System, Human Genome Center, Institute of Medical Science, University of Tokyo (https://sign.hgc.jp/signl1/index.html (accessed on 4 August 2022)). We use SiGN-L1 and estimate the azacitidine sensitivity value $m_{\alpha }$-specific gene network (α=1,…,549). For details on the kernel-based $L_{1}$-type regularization and SiGN-L1, see Shimamura et al. [39].

### 4.2. Differentially Regulated Gene Network Identification in
Azacitidine-Resistant Cell Lines

#### 4.2.1. Existing Method for Identifying Differentially Coexpressed Gene Set: DiffCoEx

Tesson et al. [40] proposed a method for identifying gene sets or clusters that are differentially co-expressed between phenotypes, called DiffCoEx. DiffCoEx measures the correlation changes of genes between two phenotypes based on the following three steps: **Step** **1.**For the drug-sensitive (-resistant) cell lines, the correlation matrix $C^{S}\left(C^{R}\right)$ is computed, where the ${\left(i,j\right)}^{th}$ entry of $C^{S}\left(C^{R}\right)$ is given as a correlation between $i^{th}$ and $j^{th}$ genes as follows,
(6)$$\begin{array}{c} C^{S}=c_{ij}^{S}=corr\left(x_{i}^{S},x_{j}^{S}\right)\;\mathrm{and}\;C^{R}=c_{ij}^{R}=corr\left(x_{i}^{R},x_{j}^{R}\right), \end{array}$$
where $x_{i}^{S}$ and $x_{j}^{S}$ ($x_{i}^{R}$ and $x_{j}^{R}$) are expression levels of the $i^{th}$ and $j^{th}$ genes in drug-sensitive (-resistant) cell lines.**Step** **2.**Compute adjacency difference matrix
(7)$$\begin{array}{c} D=d_{ij}={\left(\sqrt{\frac{1}{2}\left|\mathrm{sign}\left(c_{ij}^{S}\right){\left(c_{ij}^{S}\right)}^{2}-\mathrm{sign}\left(c_{ij}^{R}\right){\left(c_{ij}^{R}\right)}^{2}\right|}\right)}^{\gamma }, \end{array}$$
where γ is a parameter that can emphasize large correlation differences in the matrix D.**Step** **3.**Compute dissimilarity matrix T based on the topological overlap [41],
(8)$$\begin{array}{c} T=t_{ij}=1-\frac{\sum_{k}\left(d_{ik}d_{jk}\right)+d_{ij}}{\min\left(\sum_{k}d_{ik},\sum_{k}d_{jk}\right)+1-d_{ij}}. \end{array}$$The small value of $t_{ij}$ indicates that the $i^{th}$ and $j^{th}$ genes both show considerable correlation changes within the same group of genes. This implies that the groups of genes corresponding to small values of $t_{ij}$ were differentially coexpressed between phenotypes.

#### 4.2.2. Differentially Regulated Gene Network Identification from the Azacitidine Sensitivity-Specific Gene Networks

To reveal the differentially regulated gene networks between azacitidine-sensitive and -resistant cell lines, we developed a computational method for differential gene network analysis. We extended the DiffCoEx to cell line-specific gene network analysis and developed a dissimilarity measure based not on the correlation but on the edge weight in the network. Our strategy is based on the following steps in line with the DiffCoEx. 


**Step 1: Paired cell lines**
Construct paired cell lines based on randomly selected drug-sensitive and -resistant cell lines,
(9)$$\begin{array}{c} m=\{\left(m_{1}^{S},m_{1}^{R}\right),\left(m_{2}^{S},m_{2}^{R}\right),...,\left(m_{n/2}^{S},m_{n/2}^{R}\right)\}, \end{array}$$
where $m_{\alpha }^{S}$ and $m_{\alpha }^{R}$ are randomly selected $\alpha^{th}$ azacitidine sensitivity values from the drug-sensitive and -resistant cell lines, respectively. 
**Step 2: Edge weight**
From the estimated $m_{\alpha }^{S}$ and $m_{\alpha }^{S}$-specific gene networks, the edge weight $W(m_{\alpha }^{S})$ and $W(m_{\alpha }^{R})$ are computed
(10)$$\begin{array}{c} W\left(m_{\alpha }^{S}\right)=w_{ij}\left(m_{\alpha }^{S}\right)=\frac{|\beta_{ij}\left(m_{\alpha }^{S}\right)\left|+\right|\beta_{ji}\left(m_{\alpha }^{S}\right)|}{2},\\ W\left(m_{\alpha }^{R}\right)=w_{ij}\left(m_{\alpha }^{R}\right)=\frac{|\beta_{ij}\left(m_{\alpha }^{R}\right)\left|+\right|\beta_{ji}\left(m_{\alpha }^{R}\right)|}{2}, \end{array}$$
where $\beta_{ij}\left(m_{\alpha }^{S}\right)$ and $\beta_{ij}\left(m_{\alpha }^{R}\right)$ are the varying coefficients of the $j^{th}$ regulator gene on $i^{th}$ target gene in the $\alpha^{th}$ cell line. 
**Step 3: Adjacency difference matrix**
Compute the adjacency difference matrix for the $\alpha^{th}$ paired cell lines (i.e., $\left(m_{\alpha }^{S},m_{\alpha }^{R}\right)$),
(11)$$\begin{array}{c} D\left(\alpha \right)=d_{ij}\left(\alpha \right)={\left(\sqrt{\frac{1}{2}\left|\mathrm{sign}\left(w_{ij}\left(m_{\alpha }^{S}\right)\right){\left(w_{ij}\left(m_{\alpha }^{S}\right)\right)}^{2}-\mathrm{sign}\left(w_{ij}\left(m_{\alpha }^{R}\right)\right){\left(w_{ij}\left(m_{\alpha }^{R}\right)\right)}^{2}\right|}\right)}^{\gamma }, \end{array}$$
where γ is a tuning emphasizing large correlation differences in the matrix D. In practice, it is advisable to select optimal value of γ because the result heavily depends on the value of γ. 
**Step 4: Dissimilarity matrix for $\alpha^{th}$ cell line**
Compute the dissimilarity matrix T(α) for the $\alpha^{th}$ paired cell lines,
(12)$$\begin{array}{c} T\left(\alpha \right)=t_{ij}\left(\alpha \right)=1-\frac{\sum_{k}\left(d_{ik}\left(\alpha \right)d_{jk}\left(\alpha \right)\right)+d_{ij}\left(\alpha \right)}{\min\left(\sum_{k}d_{ik}\left(\alpha \right),\sum_{k}d_{jk}\left(\alpha \right)\right)+1-d_{ij}\left(\alpha \right)}. \end{array}$$Compute the average of the entries in the dissimilarity matrix
(13)$$\begin{array}{c} \mathrm{Ave}\left[T\left(\alpha \right)\right]=\frac{1}{p^{2}}\sum_{i=1}^{p}\sum_{j=1}^{p}t_{ij}\left(\alpha \right),\;\alpha =1,...,n/2. \end{array}$$
**Step 5: Dissimilarity measure of all cell lines**
Compute the dissimilarity measure of the cell line-specific gene network analysis as follows:
(14)$$\begin{array}{c} D_{CS}=\frac{1}{n/2}\sum_{\alpha =1}^{n/2}\mathrm{Ave}\left[T\left(\alpha \right)\right]. \end{array}$$
**Step 6. Statistical significance**
Shuffle cell lines into drug-sensitive and -resistant groups and construct permutation paired cell lines,
(15)$$\begin{array}{c} m^{pm}=\left\{\left(m_{1}^{S(pm)},m_{1}^{R(pm)}\right),\left(m_{2}^{S(pm)},m_{2}^{R(pm)}\right),...,\left(m_{n/2}^{S(pm)},m_{n/2}^{R(pm)}\right)\right\}. \end{array}$$For the permutation paired cell lines $m^{pm}$, Steps 2–5 are conducted, and then the dissimilarity measure $D_{CS}^{pm}$ is computed for pm=1,...,T. The permutation p value is computed as follows,
(16)$$\begin{array}{c} p.\mathrm{value}=\frac{\sum_{pm=1}^{T}I\left(D_{CS}\leq D_{CS}^{pm}\right)}{T}, \end{array}$$
where I(·) is the indicate function. 
**Step 7. Identifying differentially regulated gene networks**
For the significance level τ, we identify differentially regulated gene networks that satisfy
(17)$$\begin{array}{c} p.\mathrm{value}<\tau . \end{array}$$

The differentially regulated gene network identification between azacitidine-sensitive and -resistant cell lines was based on the parameter γ=1, significance level τ=0.05 and permutation numbers T=500.

## Figures and Tables

**Figure 1 ijms-25-03302-f001:**
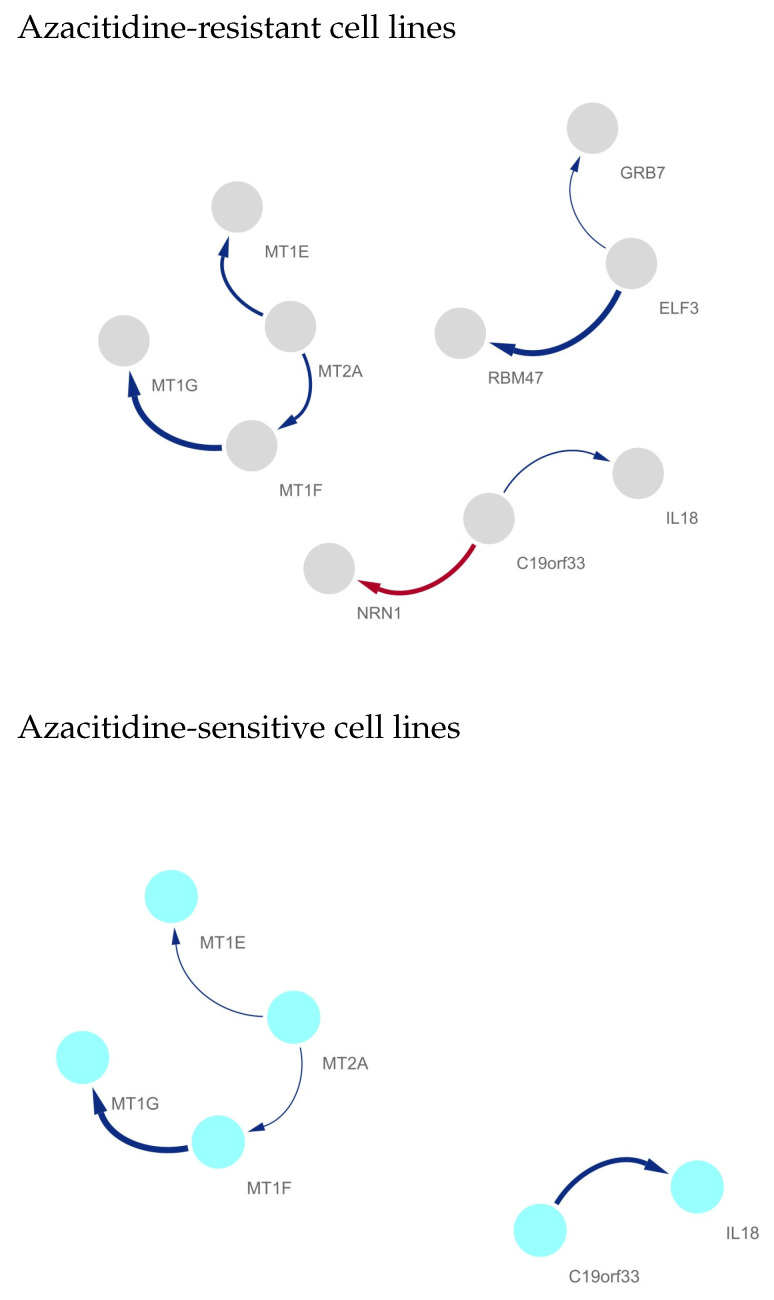
Differentially regulated gene networks between azacitidine-resistant and -sensitive cell lines, where the edge weights are given as the median of the edge strengths in 100 gene networks. The top and bottom indicate the molecular interplay in drug-resistant and -sensitive cell lines, respectively. The edge color indicates sign of the edge weight (i.e., positive—blue and negative—red), A→ B means that gene A regulates its target gene B and the thickness of line indicates the strength of edges. Azacitidine-resistant cell lines show relatively dense gene networks. Notably, the molecular interplay between GRB7, ELF3, and RBM47 exists only in drug-resistant cell lines; thus, their network can be considered to be an azacitidine resistance-specific gene network.

**Figure 2 ijms-25-03302-f002:**
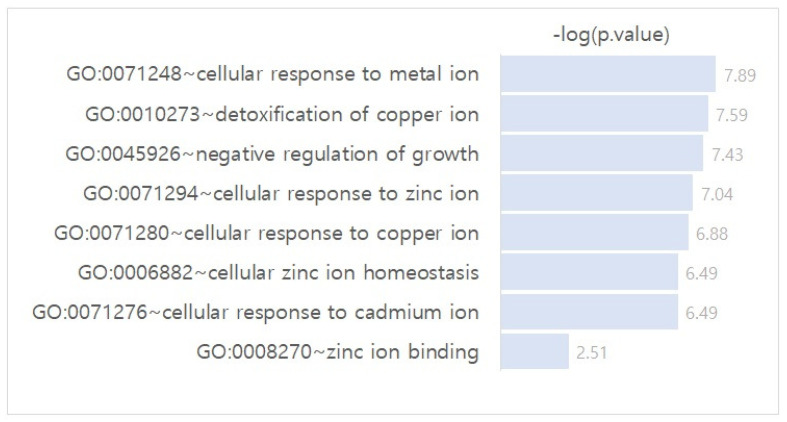
GO term pathway analysis result of the genes in the differentially regulated gene network in azacitidine-resistant cell lines.

**Figure 3 ijms-25-03302-f003:**
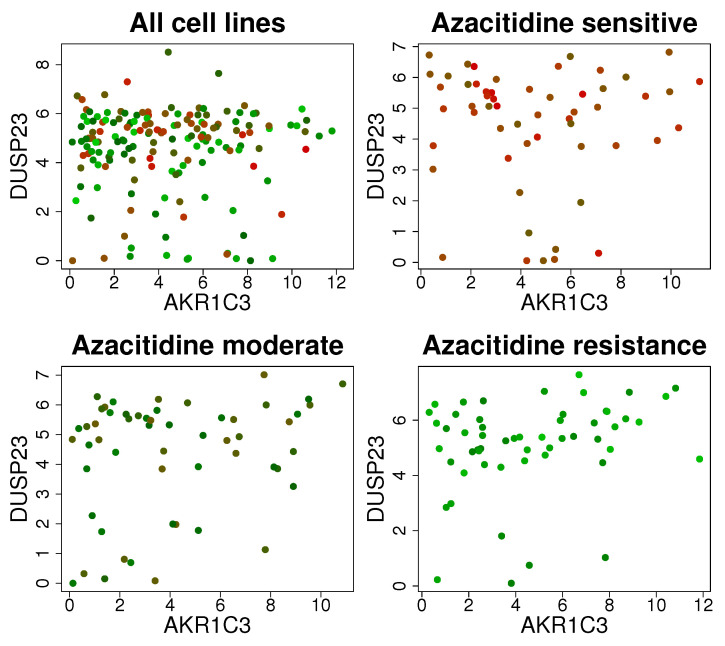
Correlation between two genes (i.e., DUSP23 and AKR1C3) under varying azacitidine sensitivities of cancer cell lines (**top left**—all cell lines, **top right**—azacitidine-sensitive cell lines, **bottom left**—moderately azacitidine-sensitive cell lines, **bottom right**—azacitidine-resistant cell lines) where the color of dots indicates drug-sensitive (red) and -resistant (green) cell lines.

**Table 1 ijms-25-03302-t001:** Identified markers from differentially regulated gene network analysis and their evidence related to AML and azacitidine.

Genes		AML		Azacitidine	
		Yes/No	Evidence	Yes/No	Evidence
C19orf33		No	-	No	-
ELF3		Yes	[17]	Yes	[18]
GRB7		No	-	No	-
IL18		Yes	[19,20,21,22,23]	No	-
Metallothionein family	MT1E	Yes	[24,25]	Yes	[26,27,28,29]
	MT1F				
	MT1G				
	MT2A				
NRN1		No	-	No	-
RBM47		Yes	[30]	Yes	[31]

## Data Availability

The datasets used in the Application section are from the Dependency Map (DepMap) projects (https://depmap.org/portal/ (accessed on 4 August 2022)).

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
