# Peer review of "Differential Gene Regulatory Network Analysis between Azacitidine-Sensitive and -Resistant Cell Lines"

_ijms, 2024, doi:10.3390/ijms25063302_

Round 1

Reviewer 1 Report

Comments and Suggestions for Authors

The manuscript entitled “Differential gene regulatory network analysis between azacitidine-sensitive and -resistant cell lines” under consideration for possible publication in International Journal of Molecular Sciences is describing an interesting study. The article is also well written. Some comments are given for authors to further improve their article.

- List of keywords needs to be modified. Authors should avoid using keywords which have appeared in the title of the manuscript.

- Better to give conclusions of the study in a single paragraph.

- A careful proofreading of the article is required.

- Figures should be self-explanatory. Authors should add more detail about figure 1 in the caption.

- References need to be updated where possible.

Author Response

We would like to thank you for insightful comments and suggestions. 
We carefully incorporated the comments and suggestions in the revised version of our manuscript. 
Our point by point responses are attached.

Reviewer 2 Report

Comments and Suggestions for Authors

Park and Miyano implement a novel mathematical technique to evaluate AML resistance to epigenetic therapies at a transcriptional level. The authors' strategy reliably found biologically-relevant markers implicated in both AML and azacytidine responses. Notably, the authors work substantially implicates metal homeostasis in AML azacytidine resistance. 

Major point:

Table 1 would benefit from reorganization. It currently appears in the text before it is mentioned and should be moved to later. While I recognize that the authors describe the references listed in Table 1 within the text, the table itself has little meaning as it's currently written. My recommendation would be to change the columns to simple yes/no columns indicating if the genes have been implicated in AML and/or Aza response (and then include the reference number). This way Table 1 can be understood at a very basic level without relying so heavily on the text from the results section. 

A few minor points: 

1. ELF3 subsection could use some brief discussion of what the referenced manuscripts claim/found 

2. The last paragraph of the introduction feels out of place and is truthfully unnecessary. I would suggest removing it. 

Author Response

(The authors gave the same response as above.)
